# Vaccinations in Selected Immune-Related Diseases Treated with Biological Drugs and JAK Inhibitors—Literature Review and Statement of Experts from Polish Dermatological Society

**DOI:** 10.3390/vaccines12010082

**Published:** 2024-01-13

**Authors:** Joanna Narbutt, Zbigniew Żuber, Aleksandra Lesiak, Natalia Bień, Jacek C. Szepietowski

**Affiliations:** 1Department of Dermatology, Pediatric Dermatology and Oncology Clinic, Medical University of Lodz, 90-419 Lodz, Poland; joanna.narbutt@umed.lodz.pl (J.N.); aleksandra.lesiak@umed.lodz.pl (A.L.); 2Department of Pediatrics, Faculty of Medicine and Health Sciences, Andrzej Frycz Modrzewski Krakow University, 30-705 Krakow, Poland; zbyszekzuber@interia.pl; 3Department of Dermatology, Venereology and Allergology, Wroclaw Medical University, 50-368 Wroclaw, Poland; jacek.szepietowski@umw.edu.pl

**Keywords:** vaccination, biological drugs, JAK inhibitors, immunization, vaccine, biologics

## Abstract

The growing use of biological drugs in immune-mediated chronic diseases has undoubtedly revolutionized their treatment. Yet, the topic of vaccinations in this group of patients still raises many concerns and implies many therapeutic problems that require discussion and standardization of management. The aim of this literature review is to present current knowledge regarding safety and efficacy of vaccinations in dermatological and rheumatological patients treated with biological drugs and JAK inhibitors. Additionally, this article provides recommendation from experts of the Polish Dermatological Society about proper use of vaccinations during therapy with biologics. Generally, all live attenuated vaccines are contraindicated during immunosuppressive/immunomodulatory therapy. If there is need, they should be administered long enough prior to the therapy or after cessation. Yet, inactivated vaccines mostly can be safely used, but the problem in this case is the effectiveness of the vaccination. Most studies report that the immune response in patients on biologics after administration of different inactivated vaccines is similar to or even better than in the control group. Thus, the importance of vaccination among patients on biologics must be emphasized to reduce omissions and the fear of possible side effects or insufficient post-vaccination response.

## 1. Introduction

A vaccine is a preparation that mimics a natural infection and leads to the development of immunity analogous to that which the body acquires during initial contact with a real microorganism (bacterium or virus). The main purpose of a vaccine is to protect against severe disease and complications that cannot be predicted. In modern medicine, the term vaccination needs to be clarified, as it is also sometimes used interchangeably to refer to cancer therapies, such as for melanoma, and specific immunotherapy in allergology. 

With regard to vaccination against infectious diseases, it is important to distinguish between active and passive protection. Active protection involves stimulating the humoral or cellular immune response, while passive protection involves the administration of antibodies. 

In order to stimulate the immune system, live vaccines containing whole, virulence-free microorganisms, inactivated vaccines containing killed viruses or bacteria or their fragments (proteins, polysaccharides), and the latest generation of vaccines containing genetic information (e.g., mRNA) on antigen production are used. 

Chronic immune or autoimmune diseases encountered in dermatology and rheumatology do not have a fully elucidated pathogenesis, so doctors, but also patients or parents, sometimes have concerns about the use of immunization in these cases. This implies many therapeutic problems, and the issue undoubtedly requires discussion and standardization of management. The most common concerns about undertaking vaccination in patients with chronic immune diseases relate to the issue of whether the vaccine used will lead to an exacerbation of the immune disease, or whether the vaccine in this case will cause adverse reactions, especially allergic reaction or intolerance. Another problem is to determine whether during the use of drugs that affect the immune system, such as methotrexate, cyclosporine, systemic corticosteroids, biological drugs, or Janus kinase inhibitors, it is possible to use immunization and, if so, in what schedule. Management of chronic diseases with immune-mediated causes is reserved for specialists in dermatology or rheumatology, while immunization is carried out by family doctors, so the lack of uniformity is often due to lack of information flow between these two groups of doctors and the lack of established management algorithms.

## 2. Material and Methods

The PubMed (URL: https://pubmed.ncbi.nlm.nih.gov, accessed on 18 December 2023) database was searched for publications in English from February 2007 until December 2023 with the usage of MesH search terms: “vaccine”, “vaccination”, “vaccination schedule”, “live-attenuated vaccination”, “inactivated vaccination”, “COVID-19 vaccine” in combination with different biological drugs and JAK inhibitors, as well as names of various dermatological and rheumatological inflammatory diseases. Further publications were searched manually from the reference list in the papers. Only full text articles were considered for the reference list. In addition, product information for each biological drug and JAK inhibitor used in article was searched for indications and contraindications for different types of vaccines. 

## 3. Safety of Different Types of Vaccines in Immunocompromised Patients or Treated with Immunosuppressive/Immunomodulatory Therapy

### 3.1. Live Attenuated Vaccines

Medical data indicate that administration of live attenuated vaccine (Table 1) to immunocompromised patients can cause serious complications, including reactivation of viruses or bacteria. There are no data at present indicating the actual risk of pathogen reactivation in patients using biologic therapies or Janus kinase inhibitors, however, current guidelines state that all live attenuated vaccines are contraindicated in these patients due to this potential risk. If there is a need to administer a live attenuated vaccine due to lack of prior vaccination or lack of evidence of immunity, it should generally be administered 14–30 days prior to therapy or at least 3 months after cessation of therapy. Therefore, an important clinical implication is that when planning immunosuppressive/immunomodulatory therapy, a thorough medical history of immunization should be taken and, if the initiation of therapy is on schedule, the applicable immunization regimen, appropriate for the patient’s age and region of residence, should be administered first [1].

### 3.2. Inactivated Vaccines

The use of inactivated vaccines (Table 1) does not carry the risk of causing infection, so they can be safely administered to patients on immunosuppressive/immunomodulatory therapy, although the problem in this case is the effectiveness of the vaccine. The Centers for Disease Control and Prevention (CDC) recommends that patients ≥ 19 years old undergoing or starting biologic therapy receive pneumococcal vaccine and annual inactivated influenza vaccine. In addition, the CDC Advisory Committee on Immunization Practices (ACIP) issued recommendations in October 2021 that all adults ≥19 years of age receiving immunosuppressive/immunomodulatory therapy should receive two doses of recombinant herpes zoster vaccine. In turn, the National Psoriasis Foundation’s 2019 recommendations suggest that all patients with psoriasis and psoriatic arthritis > 50 years of age and patients < 50 years of age on tofacitinib, combination immunosuppressive therapy, or systemic corticosteroids should receive a mandatory recombinant herpes zoster vaccine [2,3,4,5].

### 3.3. Vaccines against COVID-19

Vaccination against COVID-19 (BioNTech Manufacturing GmbH, Mainz, Germany; Moderna mRNA-1273—Moderna Biotech Spain, S.L., Madrid, Spain; Johnson & Johnson’s JNJ-78436735—Janssen Biologics B.V., Leiden, The Netherlands) can influence the course of chronic inflammatory dermatoses or even be a trigger of the new onset of those diseases. In the literature, there are several reports presenting flares or new onsets of cutaneous diseases such as lichen planus, psoriasis, atopic dermatitis, hidradenitis suppurativa, atopic eczema, or alopecia areata after COVID-19 vaccination. According to a systemic review by Martora et al., the majority of cutaneous reactions were found after mRNA vaccines, which were the most used type of COVID-19 vaccination worldwide. Moreover, the exacerbations and new onsets were reported following the first, the second, and the third dose of vaccines, highlighting that each dose may be related to their development. Yet, more studies are needed to discover the pathogenetic mechanisms related to cutaneous reactions following COVID-19 vaccination and to establish its clinical implications [6].

In 2022, the CDC published specific recommendations for COVID-19 vaccination for patients with immunodeficiencies and those receiving immunosuppressive/immunomodulatory therapy, including patients taking TNF-alpha, IL-17, IL-23, IL-12/23, and IL-4/13 inhibitors. It is now believed that patients who have received the COVID-19 mRNA vaccine (Pfizer-BioNTech BNT162b2 or Moderna mRNA-1273) should receive a third vaccination dose 28 days after the second dose, followed by an additional booster dose at least 3 months after the third dose. Patients on biologic therapy who were vaccinated against COVID-19 with Johnson & Johnson’s JNJ-78436735 vaccine should receive a second dose of mRNA vaccine against COVID-19 (Pfizer-BioNTech BNT162b2 or Moderna mRNA-1273) 28 days after the first vaccination, followed by an additional booster dose of the mRNA vaccine against COVID-19 at least 2 months after the second dose. Biologic drugs can be safely used during and after vaccination and should not be discontinued [7,8].

The rationale for such a regimen is based on the results of conducted studies, which show that double vaccination in these groups of patients is not sufficient to induce a normal immune response. Several clinical studies have shown that patients using TNF-alpha or IL-23 inhibitors produce adequate levels of antibodies, but there is no proper stimulation of the cellular response. Current data are also unable to indicate how long antibodies persist in these patients. Recently, an Italian multicenter study with psoriatic patients treated with anti-TNF-alpha, anti-IL-12/23, anti-IL-17, and anti-IL-23 has demonstrated a valid humoral response and safety profile of anti-SARS-CoV-2 vaccinations among all patients on different biologics. Ongoing observations in groups of patients not receiving immunosuppressive/immunomodulatory therapies have not shown a higher incidence of side effects following third-dose therapy, so recommendations for patients treated with biologics seem reasonable and justified [9,10,11,12,13].

## 4. Influence of Biologics on Immune System and Safety and Effectiveness of Vaccination

Biological therapies include drugs that target cytokines or receptors involved in disease development. Their use without doubt has revolutionized the treatment of immune-mediated chronic diseases such as psoriasis, psoriatic arthritis, and atopic dermatitis. These drugs, compared to conventional therapies, are also characterized by higher efficacy but, most importantly, a better safety profile [14,15,16,17,18,19]. However, there is literature evidence indicating that patients using biologics may have a higher risk of developing infectious diseases and, due to the immunosuppressive/immunomodulatory effects of these substances, have a reduced ability to mount a normal immune response [20,21].

Analysis performed by the European Society of Clinical Microbiology and Infectious Diseases concluded that use of TNF-alpha inhibitors may increase the risk of bacterial, fungal, opportunistic, and certain viral infections [22]. 

Particular attention should be paid to cases of reactivation of tuberculosis infection in patients using this treatment, hence the Quantiferon test is performed before the therapy and then once a year. Drugs from the IL-17 inhibitor group (bimekizumab, brodalumab, iksekizumab, secukinumab) are a risk factor for candidiasis, while the use of IL-23 inhibitors (guselkumab, risankizumab, tildrakizumab) is associated with a higher risk of upper respiratory tract infections [23]. A breakthrough in the treatment of atopic dermatitis is the introduction of dupilumab and tralokinumab as well as Janus kinase inhibitors (abrocitinib, upadacitinib, and baricitinib). Drugs from the latter group, i.e., upadacitinib and baricitinib, also have applications in rheumatology. The literature draws attention to the higher incidence of herpes virus infection and the risk of reactivation of herpes zoster in patients using these therapies, however, according to most data this does not apply to patients using biologic therapy [24,25,26].

The effect of each biologic therapy on the immune system is different, hence the benefit–risk index should be considered in each patient and therapies should be selected individually. With regard to vaccination, it is important to consider not only the immunosuppressive effect of treatment, but also the immunomodulatory effect, so before starting therapy, it would be advisable to review the recommended vaccination schedule and, if possible, take it into account when making therapeutic decisions which is especially important in young children [27]. 

In general, vaccinations are very well tolerated and have very few adverse reactions, but occasionally the development of rare immune reactions is observed, including hypersensitivity reactions, serum sickness, Guillain–Barré syndrome, infections, and a variety of skin manifestations, such as erythema multiforme, erythema nodosum, granuloma annulare, bullous pemphigoid, Sweet’s syndrome, Gianotti–Crosti syndrome, and cutaneous lupus [28]. In the literature, there are some reports of exacerbation of joint pain after the use of certain vaccines in patients with rheumatologic diseases, most of which were transient and of moderate course [29,30]. The extent to which these reactions develop in patients using biologic therapies or Janus kinase inhibitors is not estimated; however, it should be taken into account that the use of live attenuated vaccines in these cases may be associated with a higher risk of infection due to some degree of both immunosuppressive and immunomodulatory effects of the drugs [27]. 

Dermatologists and rheumatologists prescribing biologic therapies and Janus kinase inhibitors to patients with psoriasis, psoriatic arthritis, and AD need to have adequate knowledge on current vaccination calendars and mechanisms of action of prescribed drugs and follow vaccination recommendations in treated patients. In the article, we will present current data on this issue, facilitating practicing physicians in making therapeutic decisions.

At the same time, the authors of the article take the position that chronic dermatological and/or immune-mediated rheumatological disease is not a contraindication to vaccination, but decisions must be individualized and made on the basis of the patient’s clinical condition and the current state of medical knowledge.

## 5. Vaccinations and Biologics Used in Dermatology and Rheumatology

### 5.1. TNF-Alpha Inhibitors

The oldest of the biologic drugs, i.e., TNF-alpha inhibitors, which include adalimumab (AbbVie Deutschland GmbH Co. KG, Ludwigshafen, Germany), certolizumab (UCB Farchim SA, Bulle, Switzerland), etanercept (Pfizer Manufacturing Belgium NV, Puurs-Sint-Amands, Belgium), golimumab (Janssen Biologics B.V., Leiden, The Netherlands), and infliximab (Hospira Zagreb d.o.o., Prigorje Brdovečko, Croatia), are used to treat psoriasis and psoriatic arthritis, both in children and adults. Recently, these drugs have been increasingly used off-label to treat other diseases, such as pyoderma gangrenosum and disseminated form of granuloma annulare, hence more and more patients are the recipients of these therapies. Compared to newer biologic therapies, most information on immunization is specifically on TNF-alpha inhibitors. In studies examining the effect of drugs in this group on influenza and pneumococcal vaccination, no negative effect on immunization rates was found [30,31,32,33,34,35,36]. In some patients treated with TNF-alpha inhibitors, combination therapies are used with methotrexate or cyclosporine, among others. In these cases, vaccination recommendations are the same as for monotherapy with TNF-alpha inhibitors. Attention should be paid to the provisions of product characteristics of these drugs, where it is clearly stated that newborns born to mothers treated during pregnancy with drugs from this group must have their tuberculosis vaccination postponed (for adalimumab by 5 months, infliximab—6 months) [37]. Live vaccinations are contraindicated in the course of TNF-alpha inhibitor treatment.

### 5.2. IL-12/23 Inhibitor—Ustekinumab

**Ustekinumab** (Janssen Biologics B.V., Leiden, The Netherlands) is a monoclonal antibody directed against IL-12/IL-23, registered for the treatment of psoriasis and psoriatic arthritis. In dermatology, there are also several case reports on the use of this drug in other indications, including specific forms of ichthyosis or papular disease associated with CARD14 mutations, and these descriptions apply to both children and adults [38,39]. IL-12/23 inhibitor ustekinumab is not associated with an increase in the risk of infections [40].

In the case of Ustekinumab, live vaccinations are contraindicated, while inactivated vaccines are considered to be safe. One study showed that in patients undergoing chronic ustekinumab therapy (>3 years) and vaccination, the immune response to pneumococcal and tetanus toxoid vaccination was normal [41]. Another study conducted in a group of Crohn’s disease patients and treated with ustekinumab showed that the post-vaccination response (influenza vaccination) was higher in this group than in the control group [42].

### 5.3. IL-17 Inhibitors

A newer generation of biological drugs used in psoriasis and psoriatic arthritis are **IL-17 inhibitors**. Secukinumab (Novartis Pharma S.A.S., Huningue, France) and ixekizumab (Eli Lilly Kinsale Limited, Kinsale, Ireland) are monoclonal antibodies directed against IL-17A, bimekizumab (Rentschler Biopharma SE, Laupheim, Germany) against IL-17A/F, while brodalumab (Immunex Rhode Island Corporation, West Greenwich, RI, USA) is an antibody directed against the alpha subunit of the IL-17 receptor [43]. According to recommendations, live attenuated vaccines must not be used in patients during therapy with drugs from this group. One study has shown that the use of secukinumab does not interfere with the production of normal levels of protective antibodies after influenza and meningococcal vaccination [44,45]. Similar conclusions were reached for ixekizumab. The planned study evaluated healthy subjects who received a single dose of ixekizumab 160 mg, followed 2 weeks later by a dose of 80 mg along with vaccination against pneumococcus and tetanus. The immune response was normal, concluding that ixekizumab does not interfere with the humoral response [46]. There is information on the drug product characteristics for bimekizumab that healthy subjects who took a single 320 mg dose of bimekizumab two weeks prior to vaccination with inactivated seasonal influenza vaccine showed a similar immune response to those who did not take bimekizumab prior to vaccination [47]. In the case of brodalumab, according to the summary of product characteristics, it is recommended that patients have all vaccinations up to date according to local guidelines before starting treatment. It is also clearly emphasized that the immunization of infants with live attenuated vaccines after fetal exposure to brodalumab during the third trimester of pregnancy should be discussed with a physician [48,49]. 

### 5.4. IL-23 Inhibitors

The most recent biologic drugs used to treat psoriasis and psoriatic arthritis are **IL-23 inhibitors**, which include Risankizumab (AbbVie S.r.l., Campoverde di Aprilia, Italy), guselkumab (Biogen Inc. (BIIB), Davis Drive Research Triangle Park, SC, USA), and tildrakizumab (Samsung BioLogics Co., Ltd., Songdo Bio Way, Incheon, Republic of Korea). No data are available in the literature regarding the effect of these drugs on the safety of patients following immunization or on the ability to immunize. Thus, it is assumed that live attenuated vaccines should not be used during therapy, while inactivated vaccines are safe and can be used according to clinical need [19]. The drug characteristics of risankizumab, guselkumab, and tildrakizumab indicate that all appropriate immunizations should be considered before starting treatment, in accordance with current vaccination recommendations. Live vaccines must not be used during the use of IL-23 inhibitors. In the case of risankizumab, if a patient has received a live vaccine it is recommended to wait at least 4 weeks before starting biologic therapy, while patients treated with risankizumab should not receive live vaccines for at least 21 weeks after its termination. In the case of guselkumab, prior to live vaccine administration, therapy should be withheld for at least 12 weeks and can be resumed at least 2 weeks after vaccination. In the case of tildrakizumab treatment, therapies can be administered only 4 weeks after the administration of the live vaccine. If the patient is treated with tildrakizumab, vaccination with the live vaccine can be applied only 17 weeks after the end of treatment [50,51,52]. 

### 5.5. Dupilumab and Tralokinumab

In recent years, there has been a huge breakthrough in the treatment of moderate to severe forms of atopic dermatitis thanks to the registration of new biologic drugs and Janus kinase inhibitors. Representatives of biological drugs are **dupilumab** (Sanofi Winthrop Industrie, Le Trait, France) **and tralokinumab** (AstraZeneca Pharmaceuticals LP Frederick Manufacturing Center (FMC), Research Court, Frederick, MD, USA). Dupilumab is a human IgG4 monoclonal antibody that has the ability to block the α subunit common to the type I and II receptor complex for interleukin-4 and IL-13 while tralokinumab is a monoclonal IgG antibody against IL-13 [18,53]. These are relatively new therapies, so there are few clinical data related to vaccine safety in patients receiving these technologies. However, there has been one randomized trial that evaluated the humoral response in patients with moderate to severe AD using dupilumab 12 weeks after giving them meningococcal and tetanus vaccines [54]. There were no differences in the production of IgG class antibodies in patients using dupilumab compared to the control group, and no adverse effects were observed, indirectly indicating the safety and clinical efficacy of vaccination in AD patients using dupilumab. This study also analyzed the production of specific IgE class antibodies against vaccines and their levels were lower in patients using dupilumab compared to the control group, which in turn may indicate that AD patients using biologics have a lower risk of developing vaccine hypersensitivity. Similar effects with regard to meningococcal and tetanus IgG antibody production and safety profile were noted at 16 weeks post-vaccination in patients using tralokinumab [55]. Thus, the results of the study indicate that patients with moderate to severe atopic dermatitis using biological treatment with either dupilumab or tralokinumab can be safely vaccinated with inactivated vaccines, while, according to today’s recommendations, they should not be given live vaccines during treatment. However, there is emerging research concerning safety and effectiveness of live vaccines among patients treated with dupilumab. Wechsler et al. conducted a pre-clinical study with mice that were coadministrated with a live attenuated influenza vaccine and the dupilumab surrogate. The results are promising and suggest that IL-4Rα blockade had no impact on the efficacy of live attenuated influenza vaccine. The second part of the study examined the response to live attenuated yellow fever vaccine (YFV) in patients with asthma who recently discontinued dupilumab treatment. Gathered clinical data suggest that in the setting of therapeutic serum levels of dupilumab, the YFV was effective, demonstrated no safety concerns, and appeared to be well tolerated. However, further studies are needed to investigate the safety, tolerability, and humoral immune response to live attenuated vaccines among patients being treated with dupilumab before the changes in recommendations can be made [56].

Therefore, especially when starting these drugs in children and adolescents, there is a need to analyze the vaccination calendar and establish an individualized treatment regimen, with therapeutic vacations planned, if any. Unequivocally, the authors of the article emphasize that the disease itself, which is atopic dermatitis, especially in the period of control of the skin lesions during the use of systemic treatment, is not a contraindication to immunization, but, on the contrary, in order to prevent the aggravation of the inflammatory process of the skin, it is necessary to implement appropriate anti-infective prophylaxis.

### 5.6. Janus Kinase Inhibitors (JAK Inhibitors)

JAK inhibitors have been used in the last few years to treat rheumatologic diseases and atopic dermatitis. Among these drugs are tofacitinib (Pfizer Manufacturing Deutschland GmbH, Freiburg, Germany)—rheumatoid arthritis, psoriatic arthritis, ankylosing spondylitis, juvenile idiopathic arthritis; upadacitinib (AbbVie S.r.l., Campoverde di Aprilia, Italy)—rheumatoid arthritis, psoriatic arthritis, axial spondyloarthropathy, atopic arthritis; baricitinib (Lilly S.A., Alcobendas, Spain)—rheumatoid arthritis, atopic dermatitis, alopecia areata; abrocitinib (Pfizer Manufacturing Deutschland GmbH, Freiburg, Germany)—atopic dermatitis; and ritlecitinib (Pfizer Manufacturing Deutschland GmbH, Freiburg, Germany), recently approved in the US (alopecia areata). Ongoing clinical observations have indicated that patients on JAK inhibitors have a higher susceptibility to developing herpes zoster, hence single attempts have been made to evaluate the safety and efficacy of varicella zoster (VZV) vaccination in RA patients during JAK inhibitor therapy. The researchers used a recombinant adjuvant anti-VZV vaccine. In the study conducted, RA patients treated with JAK inhibitors underwent two vaccinations against VZV and the effects were compared with the control group. Treatment with JAK inhibitors was continued during vaccination. Among study group, eight patients were on filgotinib (Galapagos NV, Mechelen, Belgium), seven on baricitinib, six on upadacitinib, and five on tofacitinib. One month after the first administration of vaccination, the anti-VZV IgG antibody level was high and remained stable regardless of the JAK inhibitor used and, after another month, it was comparable to the group of healthy patients. No exacerbation of the underlying disease was observed. Thus, the conclusion was drawn that the immune response to VZV vaccination is not impaired during JAK inhibitor therapy [57]. Similar results were obtained by authors evaluating the efficacy and safety of anti-VZV vaccination in rheumatoid arthritis patients treated with upadacitinib [58].

The lack of many data from the literature on the issue of efficacy and safety of immunization in patients on JAK inhibitor therapy indicates the need for further studies. The summary of product characteristics of tofacitinib indicates the need for all scheduled vaccinations, especially in patients with polyarticular juvenile idiopathic arthritis and juvenile psoriatic arthritis. Live vaccines should not be administered during tofacitinib treatment; however, they can be administered in the 2–4 weeks prior to the start of treatment [59]. 

Regarding upadacitinib, baricitinib, and abrocitinib, there are no reports regarding the immune response to live vaccines, so they are not recommended during therapy with these drugs and immediately before the start of treatment. In any case, it is advisable to schedule all immunizations prior to initiating therapy with selective JAK inhibitors. This information is particularly important for the treatment of adolescents, since upadacitinib in atopic dermatitis has a registration from the age of 12. The effects of upadacitinib and baricitinib on the humoral response were evaluated after administration of an inactivated polysaccharide conjugated vaccine against pneumococcus and tetanus (for baricitinib only). The vaccine was administered to patients with rheumatoid arthritis without interrupting treatment. In both studies, the majority of patients were taking concomitant methotrexate. In both cases, pneumococcal vaccination induced a satisfactory immune response (67.5% of patients on upadacitinib 15 mg/d, 56.5% of patients on upadacitinib 30 mg/d, and 68% of patients receiving baricitinib). In 43.1% of patients on baricitinib, there was a satisfactory immune response to tetanus vaccination [60,61,62].

On the other hand, a study evaluating the effectiveness of pneumococcal vaccination in patients with rheumatoid arthritis treated with tofacitinib showed a lack of satisfactory immune response, especially in patients who were concurrently taking methotrexate. Nevertheless, discontinuation of tofacitinib therapy one week before the scheduled vaccination and reintroducing the drug one week after the vaccination significantly improved the immune response, although the difference between this group of patients and those still using the drug was not statistically significant [63]. Thus, the cited data indicate that tofacitinib has a different mechanism of action from selective JAK inhibitors.

All data on vaccinations during the treatment with ritlecitinib are from its leaflet. At the moment, there are no data on the response to vaccination in patients administered with this medication. According to the available information, use of live attenuated vaccines should be avoided during or shortly prior to initiating treatment. Prior to initiating LITFULO, it is recommended that patients should receive all vaccinations along with local immunization guidelines and prophylactic herpes zoster vaccinations prior to initiating treatment with ritlecitinib [64].

Before starting therapy with all Janus kinase inhibitors, it is additionally recommended that prophylactic vaccination against herpes zoster be administered.

### 5.7. Tocilizumab

Data in autoimmune patients treated with tocilizumab (Merck Studio S.A, Corsier-sur-Vevey, Switzerland) have shown that blocking IL-6 signaling declines IgG level, autoantibodies, memory B cells, and circulating plasma cells [65,66]. IL-6 is an important inflammatory marker predicting severity of COVID-19. A raised value of baseline IL-6 correlates with mortality, and treatment with IL-6 antagonists resulted in higher incidence of hospital survival with both tocilizumab (TCZ) and sarilumab [67]. 

Short-term TCZ treatment does not significantly attenuate humoral responses to PPV23 or TTV. To maximize vaccine response, patients should be up to date with immunizations before starting TCZ treatment. In patients with rheumatic diseases, vaccination is recommended for primary prevention of infections [68]. 

### 5.8. Rituximab

Particular attention and caution should be exercised when using drugs with a different nature of action, i.e., rituximab (RTX) anti-CD20 agent in rheumatoid arthritis (RA) and other autoimmune diseases and anifrolumab, a monoclonal antibody against the receptor for type I interferons in systemic lupus erythematosus (SLE). The use of anti-CD20 therapy, which destroys B lymphocytes, significantly weakens the response to commonly recommended vaccinations against influenza, pneumococci, and COVID-19 [69,70,71,72].

If planning the use of RTX (Genentech Inc., 1000 New Horizons Way, Vacaville, CA, USA), vaccination is recommended before starting treatment, and supplementation of vaccinations is suggested in such a way that the series of vaccines begins 4 weeks before the next planned dose of RTX. A patient receiving RTX in a two-dose cycle (2 weeks apart) with cycles repeated every 6 months is recommended to start vaccination against influenza and COVID-19 approximately 5 months after the start of the previous RTX cycle. The next administration of RTX can be scheduled for 2–4 weeks after vaccination against influenza and COVID-19 [73,74,75].

### 5.9. Anifrolumab

Anifrolumab (AstraZeneca Pharmaceuticals LP Frederick Manufacturing Center (FMC), Research Court Frederick, Maryland, MD, USA) has been registered in the EU for the treatment of moderate to severe systemic lupus erythematosus. During treatment with a type I IFN blocker, there is a risk of reactivation of herpes zoster and latent tuberculosis. Currently, there are no data on vaccinations during anifrolumab treatment in SLE patients, and it is recommended to complete vaccinations before starting therapy with this drug [76,77].

## 6. Conclusions and Future Directions

There are many reports in the literature recommending the use of immunization in patients with chronic immune mediated diseases to reduce the risk of infectious diseases [78,79,80,81,82]. 

Ideally, the patient’s vaccination history should be analyzed before starting systemic treatment and it is suggested to correct deficiencies, with regard to mandatory as well as supplementary vaccinations, such as vaccination against influenza, pneumococcus, meningococcus, Haemophilus influenzae type b, Neisseria meningitidis, hepatitis B, diphtheria, tetanus, and herpes zoster, the choice of which should depend on the patient’s immunocompetence status, age, and comorbidities [80,83]. Implementation of immunization prior to the start of systemic immunosuppressive/immunomodulatory therapy is a prerequisite for achieving an adequate immune response, i.e., a lack of influence of treatment on the post-vaccination response. Such a course of action is particularly important with live vaccines, because due to the theoretical possibility of disease development in an immunocompromised patient, they are contraindicated during immunosuppressive/immunomodulatory therapy, except when the benefit of their use far outweighs the potential risk [84]. However, this is not always possible, if only because of the need to rapidly incorporate treatment due to the severity of the underlying disease (Table 2).

We now recognize that the use of inactivated vaccines is generally safe during biologic and inhibitor therapy and is not associated with an increased risk of exacerbation of the underlying disease. 

We especially recommend the regular use of influenza and pneumococcal vaccinations in patients using biologic drugs and JAK inhibitors and vaccination against VZV in patients who will start treatment with JAK inhibitors.

The article makes it clear that vaccination should be undertaken in patients treated with biologic drugs and JAK inhibitors, as the clinical reality shows a lack of knowledge on the subject and numerous omissions. These are probably due to lack of knowledge and fear of possible side effects, as well as doubts about whether the post-vaccination response will be sufficiently effective. Thus, there is an absolute need to forge a common position between dermatology and rheumatology specialists and primary care physicians, as they are the ones who generally make decisions on immunization [85,86]. 

## Figures and Tables

**Table 1 vaccines-12-00082-t001:** Types of vaccines with examples.

Type of Vaccine	Examples
Live attenuated vaccines	The mumps, measles, rubella vaccine (MMR), oral poliomyelitis vaccine, oral typhoid fever vaccine, yellow fever vaccine (YFV).
Inactivated vaccines	Haemophilius influenzae type b, hepatitis A and B vaccine, human papillomavirus (HPV) vaccine, inactivated influenza vaccine, meningococcal vaccine, pneumococcal 13- and 23-valent vaccine (PCV13 and PPSV23), tetanus and diphtheria toxoids and acellular pertussis vaccine (TDAP), recombinant zoster vaccines (RZVs).

**Table 2 vaccines-12-00082-t002:** Safety of different types of vaccines in patients on biologics and JAK inhibitors—summary.

Drug	Live Attenuated Vaccines	Inactivated Vaccines
TNF-alpha inhibitors	Contraindicated during treatment	Generally safe *
IL-12/23 inhibitor	Contraindicated during treatment	Generally safe
IL-17 inhibitors	Contraindicated during treatment	Complete vaccination before beginning of therapy *
IL-23 inhibitors	Contraindicated during treatment	Complete vaccination before beginning of therapy *
Dupilumab and tralokinumab	Contraindicated during treatment	Generally safe
Janus kinase inhibitors (JAK inhibitors)	Complete vaccination + prophylactic vaccination against herpes zoster before beginning of therapy *	
Tocilizumab	Complete vaccination before beginning of therapy	
Rituximab	Complete vaccination before beginning of therapy *	
Anifrolumab	Insufficient data—complete vaccination before beginning of therapy	

* Detailed information in text.

## Data Availability

Not applicable.

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
