# Peer review of "Vaccinations in Selected Immune-Related Diseases Treated with Biological Drugs and JAK Inhibitors—Literature Review and Statement of Experts from Polish Dermatological Society"

_vaccines, 2024, doi:10.3390/vaccines12010082_

Round 1

Reviewer 1 Report

Comments and Suggestions for Authors

Good narrative review on a topic that concerns dermatologists and rheumatologists but also internal medicine specialists and primary care physicians. 

Abstract should provide more concise information about the work and not be just an introduction of the article. 

In lines 284-288 (reference 54), authors should discuss which JAK inhibitor/s was/were used in this study. 

More information on dupilumab use with live vaccines should be provided. 

  • DOI: 10.1016/j.jacig.2021.12.003
  •  
Comments on the Quality of English Language

English should be checked for some errors in style such as in line 315:  repeating "On the other hand" two times on a row. 

Author Response

Dear Reviewer, 

Thank you very much for your comments on our work. We changed our manuscript according to concerns from all the reviewers. Also, we added two tables to enrich the content of the article. In the attachment file you can find cover letter with all changes that have been made. We would be honored if you accept our reviewed manuscript to be published in Vaccines. 

Reviewer 2 Report

Comments and Suggestions for Authors

This important and much needed review discusses status of practical approaches in handling vaccinations in selected immune-related diseases treated with biological drugs and JAK inhibitors. Authors clearly state the aim and need for this knowledge several times in the article with concluding remarks on “an absolute need to forge a common position between dermatology and rheumatology specialists and primary care physicians, as they are the ones who generally make decisions on immunization”. This is well organized and written review highly relevant to clinicians using novel therapeutic modalities suppressing certain immune responses and subsequently recognizing possible interactions with response to vaccine(s). This review reads very well. Organized in several sections discussing 1. safety of different types (live-attenuated, inactivated and against Covid-19) of vaccines in immunocompromised patients or treated with immunosuppressive/immunomodulatory therapy followed by detailed description of specific biologics and Jak inhibitors effects on immune system and safety and effectiveness of vaccination. This review is supported with relevant references from the published literature, from the biologics manufacturers recommendations and current directives from public health institutions (CDC/ACIP). The relevance of this review is further supported by active involvement in conceptualization of this review from the experts of Polish Dermatological Society including heads of 3 leading academic dermatology and rheumatology clinics using discussed treatment modalities on daily basis. Again, well balanced opinion, important knowledge of mechanism of action and/or interaction of immune modulators with superb relevance for health practitioners.

Author Response

Dear Reviewer, 

Thank you very much for such a pleasant review of our work. We changed our manuscript according to concerns from all the reviewers. Also, we added two tables to enrich the content of the article. In the attachment file you can find cover letter with all changes that have been made. We would be honored if you accept our reviewed manuscript to be published in Vaccines. 

Reviewer 3 Report

Comments and Suggestions for Authors

Dear Authors, 

this is a very well-written and interesting manuscript, describing the interaction between vaccines and immune-modulating therapies for the most comon inflammatory skin diseases. The review is quite comprehensive and well discussed.

I have a few suggestions to further improve the manuscript:

- Even if this is a narrative review, I believe that a Materials and Methods section should be added describing how literature search was conducted

- I would discuss more deeply the impact that COVID-19 vaccines can have on inflammatory skin diseases, as described in this research, where biologics were successfully used in psoriasis patients with vaccine-induced flares (Gargiulo L, Ibba L, Vignoli CA, et al. New-onset and flares of psoriasis after COVID-19 infection or vaccination successfully treated with biologics: a case series. J Dermatolog Treat. 2023;34(1):2198050. doi:10.1080/09546634.2023.2198050)

- The Authors should differentiate the impact of vaccination in patients treated with different  drugs, such as anti.TNF alpha drugs and Interleukin inhibitors, as recently published by Sacchelli et al. ("Sacchelli L, Filippi F, Balato A, et al. PsoBioVax: A multicentric Italian case-control study of the immunological response to anti-SARS-CoV-2 vaccine among psoriatic patients under biological therapy. J Eur Acad Dermatol Venereol. Published online December 7, 2023. doi:10.1111/jdv.19662")

Comments on the Quality of English Language

English language is fine. Only a few minor typos are present in the text. No critical issues.

Author Response

Dear Reviewer, 

Thank you very much for your comments on our work. We changed our manuscript according to concerns from all the reviewers. Also, we added two tables to enrich the content of the article. In the attachment file you can find cover letter with all changes that have been made . We would be honored if you accept our reviewed manuscript to be published in Vaccines. 
